# HGOE: Hybrid External and Internal Graph Outlier Exposure for Graph Out-of-Distribution Detection

## ABSTRACT

With the progressive advancements in deep graph learning, out-of-distribution (OOD) detection for graph data has emerged as a critical challenge. While the efficacy of auxiliary datasets in enhancing OOD detection has been extensively studied for image and text data, such approaches have not yet been explored for graph data. Unlike Euclidean data, graph data exhibits greater diversity but lower robustness to perturbations, complicating the integration of outliers. To tackle these challenges, we propose the introduction of **H**ybrid External and Internal **G**raph **O**utlier **E**xposure (HGOE) to improve graph OOD detection performance. Our framework involves using realistic external graph data from various domains and synthesizing internal outliers within ID subgroups to address the poor robustness and presence of OOD samples within the ID class. Furthermore, we develop a boundary-aware OE loss that adaptively assigns weights to outliers, maximizing the use of high-quality OOD samples while minimizing the impact of low-quality ones. Our proposed HGOE framework is model-agnostic and designed to enhance the effectiveness of existing graph OOD detection models. Experimental results demonstrate that our HGOE framework can significantly improve the performance of existing OOD detection models across all 8 real datasets.

## CCS CONCEPTS

• **Computing methodologies** → **Knowledge representation and reasoning**; *Machine learning algorithms*; • **Mathematics of computing** → **Graph theory**.

## KEYWORDS

OOD Detection, Attributed Networks, Graph Neural Networks.

## 1 INTRODUCTION

Nowadays, graph-structured data have shown significant success in handling non-Euclidean relationships, prevalent in multimedia systems such as social networks [38, 56], knowledge graphs [31, 36], citation networks [6, 62], These capabilities facilitate advanced applications like scene graph generation [5, 10], fraud detection [17], and video captioning [2], by modeling complex relationships between heterogeneous data types[28, 30, 33, 39, 49, 50, 55, 58, 61]. However, a significant challenge arises from the i.i.d. assumption on which the mainstream graph learning methods depend, i.e., the

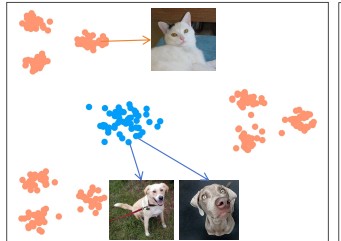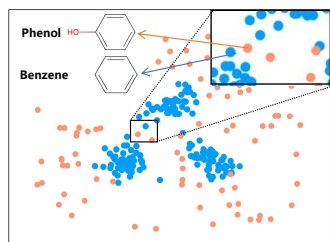

**Figure 1: Illustration of the distribution differences between images (left) and graphs (right). Blue samples are from ID classes, while orange ones are from OOD classes. Notably, ID classes form clusters in images, while in graphs, they split into subgroups with potential OOD samples in between them.**

training and testing graph data are from the same distribution. This assumption often fails in real-world scenarios, particularly in domains where data is complex and lacks sufficient labeling, such as in the case of drug molecules and proteins. For example, a new drug might not be any part of the already annotated data. Consequently, this gives rise to an interesting issue: *how to determine whether such a new drug is present in the annotated drug library?* This problem, known as Graph Out-Of-Distribution (GOOD) detection, is crucial in advancing the use of graph learning in real-world scenarios.

Although several methods for graph OOD detection have been developed [27, 32, 59], they only utilize ID graph data. When the full data distribution is complicated, merely modeling the ID data might be insufficient to capture the essential clues for OOD detection. In fact, auxiliary public OOD data is often accessible to help the detector discover such clues. Exposing such auxiliary OOD data to the model, also known as Outlier Exposure (OE), is widely studied for image data [9, 16]. However, the application of OE for graph samples has not yet been explored. To bridge this gap, we attempt to investigate the incorporation of OE into graph OOD detection.

Unlike image data, introducing outliers to assist in the OOD detection of graph data presents two significant challenges.

Firstly, since graphs are abstractions of patterns from the natural world, they exhibit inherent diversity. For instance, various non-Euclidean structures ranging from water molecules to complex social networks can be represented using graphs, yet these graphs can significantly differ in structure and properties. In contrast, images possess a Euclidean structure, which facilitates easier feature transfer. Therefore, utilizing existing graphs as outliers directly is often not effective.

Secondly, graph data exhibits less robustness to perturbations, resulting in the possibility of OOD samples existing within the boundary of ID classes. For example, as illustrated in Figure 1, phenol and benzene differ only by a hydrogen atom and an oxygen atom, yet exhibit vastly different properties; phenol is solid at room

temperature, while benzene is liquid. This suggests that slight perturbations could transform ID graph sample into an OOD sample. In contrast, image data typically exhibit a more compact intra-class distribution and more well-separated classes (*e.g.*, dogs and cats), where minor perturbations often do not change their categories. Therefore, merely incorporating some external outliers for exposure could be effective for image data. But considering that outliers may exist even within the same class of graphs, the external outlier exposure remedy is insufficient for graph OOD detection.

To address the first challenge, existing studies suggest that introducing diverse outliers closer to the in-distribution data is beneficial. Therefore, given the inherent diversity of graph data, we propose incorporating cross-domain external outliers into training. Moreover, in addition to these external outliers, we aim to synthesize outliers that are nearer to in-distribution data. Regarding the second challenge, since there exist subgroups within a class, and outliers also exist between these subgroups, we consider synthesizing some internal outliers between subgroups to assist in OOD detection. Because these samples are close to In-Distribution samples, if the model can accurately identify these samples, it would further enhance the OOD detection performance.

Motivated by this, we propose a general hybrid graph outlier exposure (HGOE) framework for graph OOD detection which integrates both external and internal outliers. The external outliers are easily collected from public database. As for internal graph outliers, we design a graphon-based ID-mixup strategy to simulate the OOD region among subgroups and synthesize OOD samples. Given these outliers, we further propose a boundary-aware OE loss to adaptively learn from true outliers and prevent the unintended bias.

In summary, the contributions of this paper are as follows:

- We propose a novel hybrid graph outlier exposure framework for graph OOD detection. It simultaneously utilizes external outliers and internal outliers to enhance the diversity. To the best of our knowledge, this is the first trial of outlier exposure in graph-level OOD detection tasks.
- To synthesize internal outliers, we design an ID-mixup method that can effectively generate outlier graph samples between ID subgroups based on graphons.
- We further introduce a novel boundary-aware loss. By instantiating the HGOE framework with a SOTA detector, we have surpassed the competitors on 8 real-world graph datasets.

## 2 RELATED WORK

**Graph Neural Networks.** Graph neural networks (GNNs) have been widely adopted in various deep learning tasks due to their ability to process graph-structured data, which can effectively extract both the structural and attribute information of graphs [12, 20, 44]. GNNs have achieved remarkable results in various deep learning tasks in recent years, such as recommendation systems, natural language processing and computer vision [46, 62]. GNNs can be broadly categorized into two distinct classes, which encompass spectral-based GNNs and spatial-based GNNs [63].

Existing Spectral-based Graph Neural Networks leverage spectral graph theory for graph analysis, offering the advantage of incorporating global graph topology information. However, they also exhibit certain limitations, including high computational complexity, challenges in handling dynamic or heterogeneous graphs, and a deficiency in local perception ability[48]. For instance, ChebNet approximates spectral graph convolutions using Chebyshev polynomials of arbitrary order, while GCN simplifies this by employing only the first two Chebyshev polynomials as the graph convolution, effectively creating a fixed low-pass filter[37].

Spatial-based Graph Neural Networks leverage the spatial information of nodes within the graph to perform graph convolutions[19]. These networks are rooted in message-passing mechanisms[20, 53], wherein each node updates its features based on the features of its neighboring nodes. Notably, Graph Attention Networks (GAT)[45] introduce a self-attention mechanism to compute node neighbor weights, allowing for dynamic and adaptive aggregation within neighborhoods. GAT further enhances model capacity by employing multi-head attention. Meanwhile, GraphSAGE[13] learns how to aggregate node features from various sources using diverse functions like mean, max-pooling, or LSTM, making it suitable for inductive learning tasks that involve new nodes or graphs during testing. Graph Isomorphism Networks (GIN)[53], another class of graph convolutional network, employ sum and MLP operations to aggregate node features. Additionally, Graph Structural Neural Networks[51] provide a versatile solution for incorporating structural graph properties into the message-passing aggregation scheme of GNNs.

**Out-of-distribution (OOD) Detection.** Existing deep learning-based classification methods often exhibit overconfidence on unseen classes [15]. To address this issue, out-of-distribution (OOD) detection [15] involves the task of distinguishing test samples from distributions different from the seen training data. It comprises post-hoc and fine-tuning approaches [54]. Post-hoc methods [21, 23, 43, 47] leverage the logit space and output scores of models that trained on in-distribution data to classify ID and OOD data. Fine-tuning approaches [9, 16] introduce extra regularization terms during training or incorporate auxiliary training data, referred to as outlier exposure, which can be either real, synthetic, or sampled from the feature space. Outlier exposure has proven effective in enhancing OOD detection performance.

However, these methods are typically applied to image or text data. OOD detection in graph data remains an under-explored area. Graph OOD detection [27] aims to determine whether test graphs originate from within the in-distribution relative to the training set or are out-of-distribution. Some studies [32, 59] focus on graph anomaly detection, where the training data comprises both in-distribution and anomalous data, positioning it as a subset of graph OOD detection. OCGIN [59] utilizes a GIN as its encoder and capitalizes on an SVDD [60] objective for one-class graph anomaly detection. GLocalKD [32] employs joint random distillation to pinpoint anomalous graphs on both local and global scales. GOOD-D [27] differentiates between ID and OOD graph data through hierarchical contrastive learning and perturbation-free graph data augmentation, having been exposed only to ID data during training. AAGOD [11] introduces a novel learnable amplifier generator, designed to generate graph-specific amplifiers, thereby enhancing the detection of OOD graphs on trained GNNs. GraphDE [22] introduces a generative framework for debiased learning and OOD detection in graph data.

**Graph Data Augmentation.** Graph data augmentation, involving transformations to enrich or enhance information in the given graph, has been extensively explored. Non-learnable augmentation methods [8, 57] achieve this through perturbation or random sampling of edges, nodes, or subgraphs. In contrast, learnable augmentation methods [7] train an Augmenter with learnable parameters using techniques like Decoupled Training, Joint Training, or Bi-level Optimization [8]. Among these methods, some utilize graph mixup [14, 25] to generate new graphs. G-mixup [14] enhances classification performance by generating new graphs through mixup based on estimating graphons of the same class. S-Mixup [25] employs soft alignments for node matching to achieve instance-level graph mixup.

As an orthogonal direction to existing graph-level OOD detection approaches, our approach focuses on introducing outlier samples that assist in OOD detection training. Our framework relies on utilizing in-distribution graphons to generate internal outliers within them, rather than directly performing augmentations on the original graph structure.

## 3 PRELIMINARIES

In this section, we provide definitions for key terms and concepts utilized throughout this paper.

### 3.1 Problem Setting

Denote a graph by $G = (\mathcal{V}, \mathcal{E}, X)$, where $\mathcal{V}$ denotes the set of nodes, $\mathcal{E}$ denotes the set of edges, and $X \in \mathbb{R}^{n \times d}$ is the node feature matrix. The adjacency matrix is denoted as $A \in \{0, 1\}^{n \times n}$, where $A_{ij} = 1$ indicates a connection between nodes $v_i$ and $v_j$ and $A_{ij} = 0$ otherwise.

In this paper, we focus on the unsupervised graph-level OOD detection problem. Specifically, given a set of unlabeled in-distribution graphs $\mathcal{D}^{in} = \{G_i\}_{i=1}^{N}$ drawn from the distribution $\mathcal{P}^{in}$, the aim of unsupervised graph-level OOD detection is to learn a graph OOD scoring function $f(\cdot)$ based on the ID data $\mathcal{D}^{in}$. A higher score $s = f(G)$ indicates a higher probability to be an OOD graph. The scoring function is evaluated on a test set $\mathcal{D}_{test} = \mathcal{D}_{test}^{in} \cup \mathcal{D}_{test}^{ood}$ ($\mathcal{D}_{test}^{in} \cap \mathcal{D}_{test}^{ood} = \emptyset$) where $\mathcal{D}_{test}^{in} \sim \mathcal{P}^{in}$ and $\mathcal{D}_{test}^{ood} \sim \mathcal{P}^{ood}$. It should be emphasized that graph data sourced from $\mathcal{P}^{in}$ and $\mathcal{P}^{out}$ might fall into multiple categories. However, in the unsupervised graph-level OOD task, the model is not provided with any category-specific labels.

### 3.2 Graphon

A graphon, denoted by the function $\mathcal{W} : [0, 1]^2 \rightarrow [0, 1]$, is a continuous, bounded, and symmetric function in graph theory, extensively used to describe graph generation [1]. The value $\mathcal{W}(i, j)$ approximates the probability of an edge between nodes $i$ and $j$ in a specific graph. A graphon can be considered as a function that embodies the characteristics of a class of graphs, allowing for the sampling of graphs from the graphon. These sampled graphs share similar topological features. Distinct graphons represent different graph classes, enabling comparative analysis of their structural features. These graphons are estimated using step function approximations [29]. Graphons are particularly useful in synthesizing new

graphs that reflect patterns found in real-world data, offering insights into the underlying structure of complex networks. However, there is no closed-form expression for graphons. Previous studies [14, 52] have employed a two-dimensional step function to estimate graphons, which can be considered a matrix representing the probability of edge existence. In this paper, we denote it as $W \in [0, 1]^{D \times D}$, where $D$ is the dimensionality of the graphon. This matrix $W$ can then be used to generate graphs with a number of nodes less than $D$.

## 4 METHODOLOGY

### 4.1 Overall Framework

In general, with a graph OOD score function $f(G)$, the basic learning objective for unsupervised graph OOD detection can be described as:

$$\min_f \mathbb{E}_{G \sim \mathcal{D}_{in}}[\mathcal{L}_{ood}(f(G))], \quad (1)$$

where $\mathcal{L}_{ood}$ is the loss function.

By integrating the hybrid graph outlier exposure (HGOE) procedure, we hope the exposed OOD data have higher OOD scores than ID data. Subsequently, we simultaneously minimize the OOD score of ID data and maximize that of OOD data:

$$\min_f \mathbb{E}_{G \sim \mathcal{D}_{in}} [\mathcal{L}_{ood}(f(G)) + \\ \beta \cdot \mathbb{E}_{G' \sim \mathcal{D}_{OE}}[\mathcal{L}_{GOE}(f(G), f(G'))]]. \quad (2)$$

Apparently, such an introduction of HGOE is independent of the basic graph OOD model and thus can be applied to most existing models.

In deploying the GOE framework, we identify two pivotal challenges: **(C1)** How to acquire high-quality graph outlier data? **(C2)** How to design a $\mathcal{L}_{GOE}$ to effectively utilize these OE data? To tackle **(C1)**, we introduce both internal and external outliers. To address **(C2)**, we develop a boundary-aware loss function. The overall pipeline of HGOE is illustrated in Figure 2. To facilitate the generation of internal outliers, we start with the division of graphs into distinct subgroups, wherein each subgroup is defined by the similarity in properties among its constituent graphs. This is followed by the estimation of the graphon for each subgroup. Thereafter, we deploy an ID-mixup technique that combines the graphons from these subgroups to obtain internal outliers, incorporating node feature information from external outliers into this process. The resulted internal and external outliers are then fed into the OOD detector, which is further optimized through the implementation of an innovatively designed boundary-aware loss. In subsequent sections, we will introduce the HGOE framework in detail.

### 4.2 Mixture Outlier Training Strategy

Our mixture outlier training strategy incorporates outliers from two sources: real-world external outliers and synthesized internal outliers. External outliers are derived from public graph datasets similar to the training graphs, while internal outliers are generated from ID graphs.

**External Outliers.** In contrast to image data, graph data in the real world display significantly more complex structures, with notable gaps in characteristics between fields like social networks and protein networks. To investigate the role of real-world outliers,

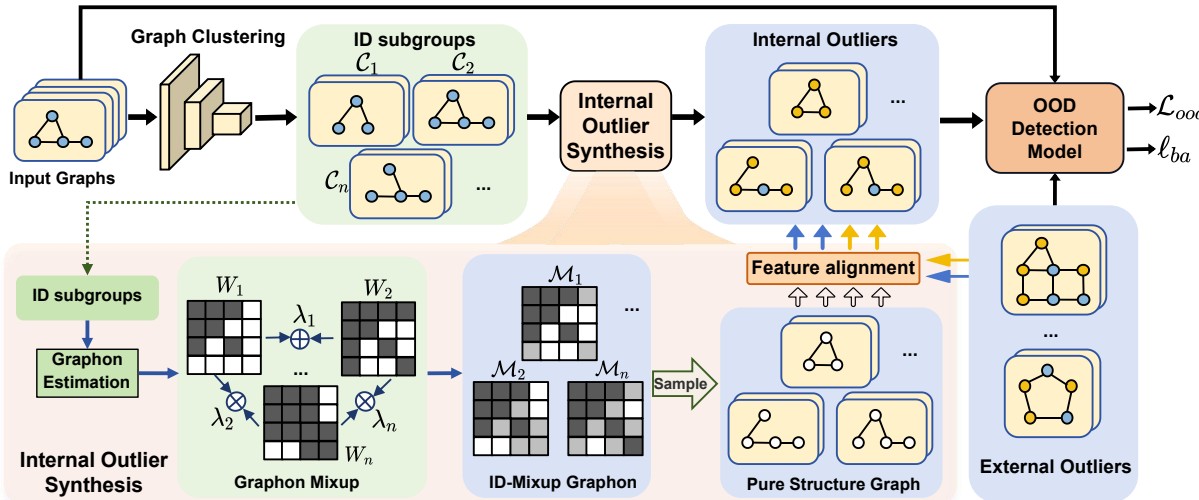

Figure 2: Overview of the proposed HGOE framework. We collect external outliers via public graph database. Given ID graphs, we perform feature extraction using GraphCL and cluster them to obtain multiple ID subgroups. Then we estimate graphons for these subgroups and mix them up to obtain graphons for internal outliers. Sampled graph structures are further used to generate node features by aligning features from external outliers. Finally, the synthesized internal outliers and real-world external outliers are jointly optimized using a boundary-aware loss.

we sample from several graph datasets to create a real-world OE dataset. During the training of different ID data, we select graphs from the OE dataset that are consistent with them in feature dimensions, ensuring no overlap between the OE graphs and the training and test sets. This process results in an external outlier dataset, denoted as $\mathcal{D}_{oe}^{ext}$.

**Internal Outliers.** The quality and diversity of external outliers data are crucial for outlier exposure, but are limited by the scope and quality of the existing auxiliary dataset. Synthesizing additional outliers then emerges as a solution, offering tailored, abundant, and diverse graph data for training. However, the challenge lies in ensuring that the synthesized data resonates with real-world outlier scenarios and does not introduce unintended biases. As discussed in the introduction, there exist internal outliers among the in-distribution graph data, which are distributed near the boundaries of subgroups. To obtain these internal outlier samples, we design an internal outlier generation strategy, which will be described in the next subsection.

### 4.3 Internal Outlier Synthesis

A straightforward way to generate graphs is to interpolate each pair of samples in the dataset directly. However, unlike Euclidean data such as images, different graphs usually have different sizes (*i.e.*, different numbers of nodes) and have unique topology in Non-Euclidean spaces. Therefore, we propose to mix up the graph generators instead of graph samples themselves to synthesize more realistic graph data.

To find internal outliers for input graphs, it is imperative to first divide the graphs into subgroups. For the input ID graphs, we adopt graph contrastive learning techniques (*e.g.*, GraphCL [57]) for feature extraction, and then perform $k$-means clustering to

obtain $k$ subgroups $C_i$, where $\mathcal{D}^{in} = \bigcup_{i=1}^{k} C_i$. Since the members in each subgroup often share similar properties, we assume that each subgroup within the ID data is generated by a specific graphon. And we use the widely-used universal singular value thresholding (USVT) method [4] to estimate the graphon $W_i$ for each $C_i$ in $\mathcal{D}^{in}$.

For each pair of non-overlapping subgroups $C_i$ and $C_j$ in $\mathcal{D}^{in}$, we perform a mixup operation over the corresponding graphons $W_i$ and $W_j$ as follows:

$$\mathcal{M} = \lambda W_i + (1 - \lambda) W_j, \qquad (3)$$

where $\lambda \in [0, 1]$ is the balancing hyperparameter. The mixup result $\mathcal{M}$ combines the structural features of subgroups $C_i$ and $C_j$, and can be considered a new graph generator positioned between the two subgroups. By sampling based on this graphon, we could obtain graphs lying in the interpolated regions between these subgroups, preserving the subgroups's topologies.

**Random Size Based Sampling.** The interpolated graphon $\mathcal{M} \in [0, 1]^{N \times N}$ has the capability to generate infinitely many graphs. However, the naive generated graphs are very likely to have a size around $N$. This limits the diversity of the synthesized outliers, which does not meet our goal in the HGOE framework to synthesize outliers broadly distributed near the original subgroups, with various sizes of graphs. In order to increase the diversity of outliers, we employ a random size based sampling strategy, *i.e.*, we first randomly sample the size of the graph $r \in [2, N]$ and then generate the graph from the sampled graphon $\mathcal{M}' \in [0, 1]^{r \times r}$. The existence of an edge between nodes $i$ and $j$ is determined by sampling from a Bernoulli distribution with the parameter $\mathcal{M}'(i, j)$.

**External Feature Alignment.** Merely sampling from the interpolated graphons could only produce pure structure graph outliers with the structure information (denoted as $A'$). Therefore, we further propose to generate node features based on external outlier

features. Specifically, we calculate the structural features of both the generated internal outlier structure $A'$ and the external outliers. And then we construct the node features for the generated internal outliers based on that of the external outliers with the most similar structure features. Here, the structural features $s_i^{\text{diff}}$ of the $i$-th node in a $d_s$-dimensional space is obtained through a $d_s$-step random walk-based diffusion process on the graph:

$$s_i^{\text{diff}} = \left[ T_{ii}, T_{ii}^2, \ldots, T_{ii}^{d_s} \right] \in \mathbb{R}^{d_s}, \tag{4}$$

where $T = A'D^{-1}$ denotes the transition matrix for the random walk on the graph, and $D$ is the degree matrix of $A'$. After computing the structural features of $A'$, we compare them with the structural features of external outliers. For each node in the synthesized internal outlier $G'$, we assign node features that are most closely aligned with the structural features of nodes in the external outliers. Note that our external feature alignment approach is training-free. By performing structural searches on nodes with existing features, we can effectively assign outlier features to nodes that initially lack features.

Through the aforementioned ID-mixup based procedure, we can generate an arbitrary number of internal outliers, denoted as $\mathcal{D}_{oe}^{int}$. Compared to real-world outliers, these synthetic outliers are not limited in quantity or source. Furthermore, being located near in-distribution samples, they effectively compensate for the uncontrolled quality issue of external outliers.

## 4.4 Boundary-Aware OE Loss

The introduced outliers further present two critical issues: How can we ensure that the introduced outliers do not fall within the in-distribution area, and how to find those more important outliers? Take the social graph ID data as an example. Intermixing social graphs could still result in a social graph, so the generated samples are not OOD data and should be excluded from the outlier dataset. At the same time, different outliers can have varying levels of importance, with points on the boundary potentially being critical points where a change in properties is about to occur.

To solve these problems, we design the following boundary-aware OE Loss $\ell_{ba}$ which is adaptively aware of whether the sample is in ID or OOD space. For an input outlier graph $G'$, the loss is calculated as:

$$\ell_{ba}(s_{G'}, \tau) = -(l - s_{G'})^\gamma \max(\log(s_{G'}), \tau), \tag{5}$$

where $s_{G'} = \text{sigmoid}(f(G'))$ is the OOD score scaled by a sigmoid function, $l$ and $\gamma$ are hyperparameters. Note that $\tau$ is an ID-boundary threshold, which we adaptively set as the smallest $s$ among the ID samples:

$$\tau = \min_{G \in \mathcal{D}_{in}} \text{sigmoid}(f(G)). \tag{6}$$

To better understand Eq. (5), we rewrite it in the following form:

$$\ell_{ba}(s_{G'}, \tau) = \begin{cases} -(l - s_{G'})^\gamma \log(s_{G'}), & \text{if } \log(s_{G'}) > \tau \\ -\tau(l - s_{G'})^\gamma, & \text{if } \log(s_{G'}) \leq \tau \end{cases}. \tag{7}$$

From this formulation, we have the following observations:

- When $\log(s_{G'}) > \tau$, it indicates that $G'$ is a valid outlier. In this case, the smaller $s_{G'}$ is, the closer it is to the in-distribution boundary. Apparently, such near-boundary outliers are important in guiding the detector to distinguish OOD samples from ID ones. Therefore, we weigh it by the term $(l - s_{G'})^\gamma$, giving higher weights to these boundary samples. As $\gamma$ increases, the model pays more attention to samples nearer to the ID boundary.
- When $\log(s_{G'}) < \tau$, it means that the outlier $G'$ is very likely invalid and fall into in-distribution space. Blindly treating it as an outlier might be harmful to the learning. Therefore, by minimizing $-\tau(l - s_{G'})^\gamma$, the OOD score is reduced, which enhances the recognition ability for in-distribution samples.

Then, considering both external and internal outliers, the total graph outlier exposure loss is formulated as:

$$\mathcal{L}_{\text{GOE}} = \sum_{G' \in \mathcal{D}_{OE}^{int} \cup \mathcal{D}_{OE}^{ext}} \ell_{ba}(s_{G'}, \tau). \tag{8}$$

Finally, we can instantiate the overall training objective $\mathcal{L}$ as follows:

$$\begin{aligned} \mathcal{L} &= \sum_{G \in \mathcal{D}_{in}} \left[ \mathcal{L}_{\text{ood}}(f(G)) + \beta \sum_{G' \in \mathcal{D}_{OE}} \ell_{ba}(s_{G'}, \tau) \right] \\ &= \sum_{G \in \mathcal{D}_{in}} \mathcal{L}_{\text{ood}}(f(G)) + \beta \sum_{G' \in \mathcal{D}_{OE}^{int} \cup \mathcal{D}_{OE}^{ext}} \ell_{ba}(s_{G'}, \tau). \end{aligned} \tag{9}$$

In practice, our $\mathcal{L}_{\text{GOE}}$ can be added as a regularization term to most existing OOD detection models, without modifying their network architectures.

## 5 EXPERIMENTS

### 5.1 Experimental Setup

*5.1.1 Datasets.* We select three pairs of datasets from the TU dataset [35] and five pairs from the OGB dataset [18]. Each pair of datasets belongs to the same field and shares similar features, but exhibits distribution shifts between the two datasets in the pair. 90% of the In-Distribution (ID) samples are used for training, while 10% of the ID samples and an equivalent number of OOD samples are used for testing. For more detailed information about these datasets, refer to Table 1.

For external outlier data, we grouped datasets with identical feature counts into a unified external dataset. Based on the feature counts of our selected datasets, as detailed in Table 1, this organization resulted in two distinct external dataset collections, with node feature numbers being 1 and 9, respectively. In this configuration, for each In-Distribution dataset, we utilized other datasets from the corresponding external dataset collection for training purposes, deliberately excluding both the OOD dataset for testing and the ID dataset itself.

*5.1.2 Competitors.* We adopt the following three categories of graph OOD detection methods as our competitors:

- **Non-deep Two-step Methods.** We use WL [41] graph kernels as feature extractors and employ local outlier factor (LOF) [3], one-class SVM (OCSVM) [34], and isolation forest (iF) [26] as detectors to perform OOD detection.

**Table 1: Statistics of datasets**

| Dataset | #Feature | #Graphs | #Avg.Nodes | #Avg.Edges | #Avg.deg |
|---------|----------|---------|------------|------------|----------|
| AIDS | 1 | 2000 | 15.69 | 16.2 | 1.03 |
| DHFR | 1 | 756 | 42.43 | 44.54 | 1.05 |
| ENZYMES | 1 | 600 | 32.63 | 62.13 | 1.90 |
| PROTEIN | 1 | 1113 | 39.06 | 72.82 | 1.86 |
| IMDB-M | 1 | 1500 | 13 | 65.93 | 5.07 |
| IMDB-B | 1 | 1000 | 19.77 | 96.53 | 4.88 |
| Tox21 | 9 | 7831 | 18.57 | 19.29 | 1.04 |
| SIDER | 9 | 1427 | 33.64 | 35.35 | 1.05 |
| FreeSolv | 9 | 642 | 8.72 | 8.38 | 0.96 |
| ToxCast | 9 | 8576 | 18.78 | 19.26 | 1.03 |
| BBBP | 9 | 2039 | 24.06 | 25.95 | 1.08 |
| BACE | 9 | 1513 | 34.08 | 36.85 | 1.08 |
| ClinTox | 9 | 1477 | 26.15 | 27.88 | 1.07 |
| LIPO | 9 | 4200 | 27.04 | 29.49 | 1.09 |
| Esol | 9 | 1128 | 13.28 | 13.67 | 1.03 |
| MUV | 9 | 93087 | 24.23 | 26.27 | 1.08 |
| BZR | 1 | 405 | 35.75 | 38.36 | 1.07 |
| COX2 | 1 | 467 | 41.22 | 43.45 | 1.05 |
| PTC_MR | 1 | 344 | 14.28 | 14.69 | 1.03 |
| MUTAG | 1 | 188 | 17.93 | 19.79 | 1.10 |

- **Deep Two-step Methods.** The process is similar to the above methods, but the feature extractor is replaced with graph deep self-supervised methods InfoGraph [42] and GraphCL [57], and the detectors are replaced with isolation forest (iF) and Mahalanobis distance (MD) [40]. Compared to graph kernels, self-supervised methods can extract features of graphs better.
- **End-to-end Methods.** We utilize three popular end-to-end learning methods, including OCGIN [59], which uses graph neural networks for feature extraction and is optimized with SVDD. GLocalKD [32] uses distillation learning for graph anomaly detection, and GOOD-D [27] employs multi-level contrastive learning for end-to-end OOD detection.

For the proposed framework, we instantiate it on the SOTA graph OOD detector baseline GOOD-D [27]. Besides, we also implement two ablated variants to show the impact of internal and external outliers. Specifically, HGOE *w/o IO* and HGOE *w/o EO* denote the variants without internal and external outliers, respectively.

*5.1.3 Implementation Details.* For HGOE, the ratio of external to internal outliers was set to 1:1, with the total number equal to the number of in-distribution samples seen during training. The hyperparameter $\lambda$ was set to 2, and $\gamma$ was set to 2. The dimension of the structural features $s_i^{\text{diff}}$ was set to 16, and the number of clusters for all datasets was determined to be 3. GraphCL was run for 50 iterations with an embedding dimension of 32, and the probabilities for node dropping, feature masking, and edge removing were all set to 0.1. During ID-mixup, $\lambda$ was randomly chosen from the range [0.01, 1]. AUC (Area under the ROC Curve) [24] is used as the performance metric. A higher AUC value indicates better detection performance.

## 5.2 Main Results

We evaluate the performance of HGOE and other competitors, with the results presented in Table 2. Our findings include:

**(1)** Two-step detection methods do not perform as well as end-to-end detection methods, especially where non-deep learning methods are inferior to deep learning-based feature extraction methods. Within self-supervised methods, GraphCL-MD demonstrates the best performance, showing GraphCL's capability in extracting features for graph OOD detection. This also explains why we chose to utilize GraphCL features for clustering.

**(2)** For our HGOE framework, compared to GOOD-D without the use of graph outliers, there was an enhancement in performance across all 7 datasets. On the ENZYMES+PROTEIN dataset, the average performance improve from 60.15 to 64.44. On Tox21+SIDER, it increased from 64.98 to 68.24, and on FreeSolv+ToxCast, the performance increase from 78.79 to 83.36.

**(3)** The performance improvement on IMDB-M+IMDB-B was not substantial because both are social datasets, and the outliers we introduce were from molecular and protein datasets, which are biologically oriented. However, using these as outliers still contributed to enhanced detection on IMDB-M+IMDB-B. This indicates that selecting outliers similar to the ID distribution, as opposed to introducing outliers completely unrelated to the ID data, is more effective.

## 5.3 Visualization

*5.3.1 ID-mixup Visualization.* After obtaining the subgroups within the known distribution, we performed a mixup of the estimated graphons and visualized the resulting mixed graphons in the form of heatmaps, as shown in Figure 3. The graphon in the center represents the result of performing ID-mixup on the two adjacent graphons. It is evident that the graphon after ID-mixup retains the structures of the original graphons, forming a new graph generator. This demonstrates that our ID-mixup can effectively blend the distributions of subgroups to a certain extent, fulfilling our hypothesis.

*5.3.2 Score Distribution Visualization.* Based on the OOD scores for samples in the test set, we visualize the frequency distribution of OOD scores for both ID and OOD samples using different colors. As shown in Figure 4, the left column depicts the score distributions without using HGOE, while the right column shows the results after applying the HGOE framework. The less the overlap and the greater the distance between the ID and OOD areas are, the better the corresponding model's performance is. It is observable that our HGOE method has decreased the overlap area between ID and OOD distributions, which also explains the reason for the performance improvement.

*5.3.3 Visualization of Graph-level Features.* We extract features of each individual graph by the OOD detection model. These extracted features are then visualized through t-SNE, as demonstrated in Figure 5. The representation employed in this study is the graph-level representation acquired from HGOE. It can be observed that, apart from the visualization result of AIDS+DHFR in 5a, the representations of other OOD graphs are distributed among the ID graphs. This highlights the importance of identifying and analyzing internal outliers that reside within the distribution of ID graphs for more effective and accurate OOD Detection.

Table 2: AUC (mean±std%) results on seven real-world graph datasets. The **best** and **runner-up** performances are highlighted.

| ID dataset
OOD dataset | AIDS
DHFR | ENZYMES
PROTEIN | IMDB-M
IMDB-B | Tox21
SIDER | FreeSolv
ToxCast | BBBP
BACE | ClinTox
LIPO | Esol
MUV |
|---|---|---|---|---|---|---|---|---|
| WL-LOF | 50.77±2.87 | 52.66±2.47 | 52.28±4.50 | 51.92±1.58 | 51.47±4.23 | 52.80±1.91 | 51.29±3.40 | 51.26±1.31 |
| WL-OCSVM | 50.98±2.71 | 51.77±2.21 | 51.38±2.39 | 51.08±1.46 | 50.38±3.81 | 52.85±2.00 | 50.77±3.69 | 50.97±1.65 |
| WL-iF | 50.10±0.44 | 51.17±2.01 | 51.07±2.25 | 50.25±0.96 | 52.60±2.38 | 50.78±0.75 | 50.41±2.17 | 50.61±1.96 |
| InfoGraph-iF | 93.10±1.35 | 60.00±1.83 | 58.73±1.96 | 56.28±0.81 | 56.92±1.69 | 53.68±2.90 | 48.51±1.87 | 54.16±5.14 |
| InfoGraph-MD | 69.02±11.67 | 55.25±3.51 | **81.38±1.14** | 59.97±2.06 | 58.05±5.46 | 70.49±4.63 | 48.12±5.72 | 77.57±1.69 |
| GraphCL-iF | 92.90±1.21 | 61.33±2.27 | 59.67±1.65 | 56.81±0.97 | 55.55±2.71 | 59.41±3.58 | 47.84±0.92 | 62.12±4.01 |
| GraphCL-MD | 93.75±2.13 | 52.87±6.11 | 79.09±2.73 | 58.30±1.52 | 60.31±5.24 | 75.72±1.54 | 51.58±3.64 | 78.73±1.40 |
| OCGIN | 86.01±6.59 | 57.65±2.96 | 67.93±3.86 | 46.09±1.66 | 59.60±4.78 | 61.21±8.12 | 49.13±4.13 | 54.04±5.50 |
| GLocalKD | 93.67±1.24 | 57.18±2.03 | 78.25±4.35 | 66.28±0.98 | 64.82±3.31 | 73.15±1.26 | 55.71±3.81 | 86.83±2.35 |
| GOOD-D | 98.70±0.82 | 60.15±0.46 | 79.10±1.32 | 64.98±0.42 | 78.79±4.22 | 80.60±2.60 | 67.41±3.38 | 90.52±1.54 |
| HGOE *w/o IO* | 99.14±0.43 | **62.17±1.44** | 78.56±1.44 | **66.66±2.02** | **83.36±1.35** | 81.52±0.91 | **70.78±1.84** | **92.47±2.29** |
| HGOE *w/o EO* | 98.82±0.21 | 62.12±1.42 | 80.04±1.63 | 65.49±1.33 | 82.73±2.99 | 81.42±2.83 | 68.63±2.95 | 91.37±1.69 |
| HGOE | **99.28±0.34** | **64.44±2.19** | 81.74±2.25 | **68.24±0.60** | **82.89±2.33** | **83.46±1.79** | 70.09±1.52 | **92.64±2.44** |

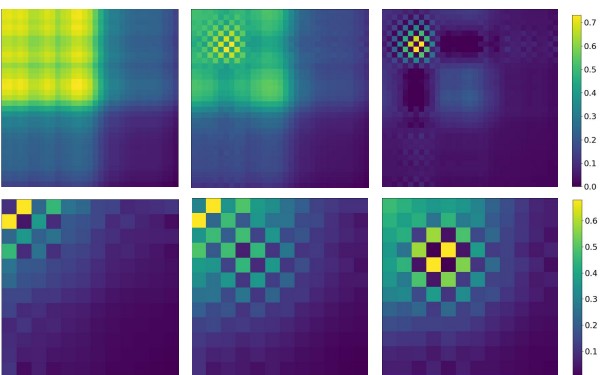

Figure 3: Visualization of graphons obtained by ID-mixup. The rows are from ENZYMES and FreeSolv datasets, respectively. The first and third columns are graphons of two subgroups, and the second column shows their mixup results. Brighter cells indicate a higher probability of edge existence at that location.

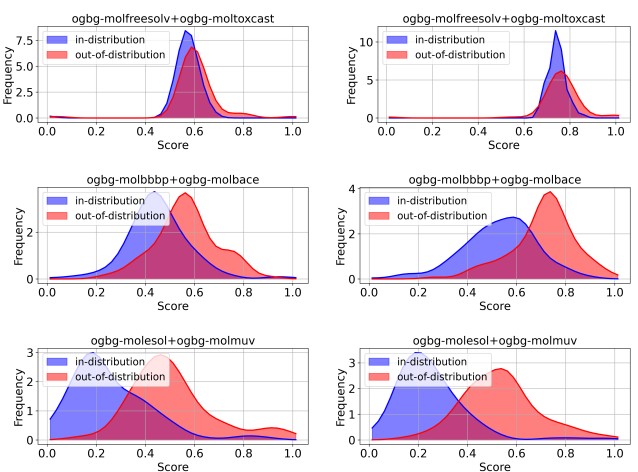

Figure 4: Score distributions on several graph datasets. The left column shows results without HGOE, while the right column is with HGOE. It is evident that the overlap area between ID and OOD samples becomes smaller after introducing HGOE.

## 5.4 Ablation Study

*5.4.1 Using Only One Type of Outliers.* In Table 2, HGOE *w/o IO* and HGOE *w/o EO* represent the results of using only internal outliers and external outliers, respectively. We observe that using just one of these types yields better results than not using any OE (Outlier Exposure) samples at all. Moreover, in most cases, the performance of using only external outliers is superior to that of using only internal outliers, but inferior to using both types. This indicates that the combination of both types of outliers can more effectively enhance the performance of hybrid graph outlier exposure.

*5.4.2 The Strategy for Adaptive Allocation of $\tau$.* In Section 4.4, we introduce an adaptive parameter $\tau$ in the distribution-aware OE loss, which is set as the smallest normalized score among the ID samples. To explore the specific role of $\tau$, we implement methods setting $\tau$ according to the maximum, average, and minimum normalized scores. The results, as shown in Figure 3, indicate that the min strategy for allocating $\tau$ performs the best. However, this strategy negatively affects performance on the IMDB-M+IMDB-B dataset, which we believe is due to its significant difference from the other datasets.

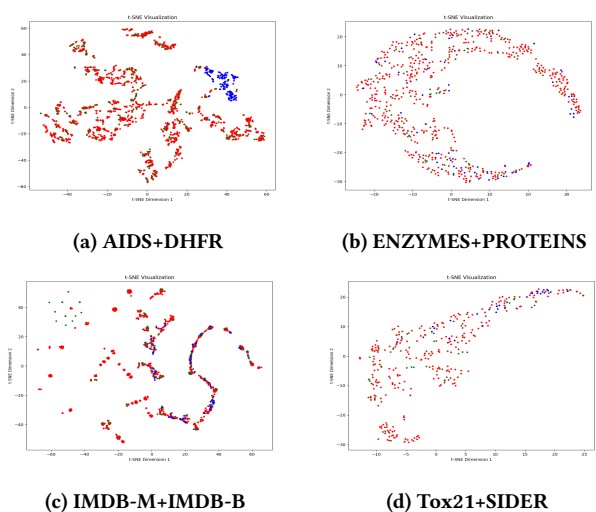

(a) AIDS+DHFR      (b) ENZYMES+PROTEINS

(c) IMDB-M+IMDB-B      (d) Tox21+SIDER

**Figure 5: t-SNE visualization of graph-level representations.
Red dots represent ID graphs in the training set, green and
blue dots represent ID and OOD graphs in the test set, respectively. It can be observed that OOD samples also exist within
ID classes.**

**Table 3: Performance gains of different methods for choosing
$\tau$ relative to not using the threshold $\tau$.**

|      | AIDS DHFR | ENZYMES PROTEIN | IMDB-M IMDB-B | Tox21 SIDER | FreeSolv ToxCast | BBBP BACE | ClinTox LIPO | Esol MUV |
|------|-----------|-----------------|---------------|-------------|------------------|-----------|--------------|----------|
| min  | +0.09 | **+0.38** | -0.34 | **+0.23** | **+0.62** | **+0.20** | **+0.31** | **+0.45** |
| mean | **+0.19** | +0.24 | **-0.17** | +0.03 | +0.44 | -0.31 | +0.02 | -0.30 |
| max  | -0.03 | -0.38 | -0.26 | -0.46 | -0.78 | -1.26 | -0.12 | -0.43 |

**Table 4: Performance of HGOE with only internal outliers at
different $\lambda$ ranges.**

| $\lambda$ Range | ENZYMES PROTEIN | Tox21 SIDER | FreeSolv ToxCast | BBBP BACE | ClinTox LIPO |
|-----------------|-----------------|-------------|------------------|-----------|--------------|
| [0.01, 1.0] | **62.12±1.42** | **65.49±1.33** | **82.73±2.99** | **81.42±2.83** | 68.63±1.33 |
| [0.1, 0.9] | 62.07±1.82 | 65.20±1.58 | 82.06±2.23 | 81.01±3.21 | 68.20±1.58 |
| [0.3, 0.7] | 62.03±1.29 | 65.13±1.41 | 81.92±2.26 | 81.20±2.78 | **68.76±2.65** |
| [0.4, 0.6] | 61.98±1.13 | 65.29±1.34 | 81.92±1.87 | 81.34±2.87 | 68.74±3.00 |

## 5.5 Sensitivity Analysis

*5.5.1 ID-mixup Weight $\lambda$.* We explore the sensitivity of HGOE with
respect to $\lambda$ by evaluating its performance during ID-mixup across
different $\lambda$ ranges. As seen in Table 4, we progressively narrow the
range of $\lambda$ from [0.01, 1.0] to [0.4, 0.6]. In this process, a performance decline was observed in most datasets. This suggests that for
ID-mixup, blending two in-distribution samples at varying ratios
can increase the diversity of internal outliers, thereby enhancing
detection effectiveness. However, an opposite trend of performance
increase, rather than decrease, was observed in the CLintox+LIPO

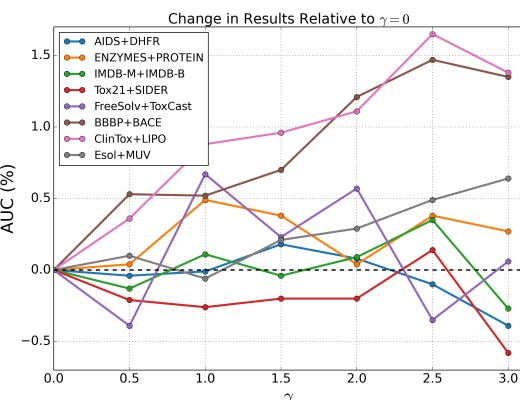

**Figure 6: Performance gain of HGOE compared to $\gamma = 0$ when
$\gamma$ varies.**

dataset pair. This anomaly may be related to the characteristics
of these datasets, both being molecular graphs from the OGB and
possessing extremely similar average node and edge counts, resulting in closer distributions. If ID-mixup outliers falling within
the in-distribution range, it could lead to a greater performance
decrease compare to other datasets. So when $\lambda$ is set to a middle
value, the generated graphon lies between subgroups. This explains
why choosing a $\lambda$ value range [0.3, 0.7] yields better results.

*5.5.2 Hyperparameter $\gamma$ of Boundary-aware OE Loss.* As described
in our framework, $\gamma$ plays a significant role in the boundary-aware
loss. We vary the value of $\gamma$ to {0, 0.5, 1.0, 1.5, 2.0, 2.5, 3.0}, and
as observed in Figure 6, as $\gamma$ increases, there is a performance
improvement when $\gamma \in [0, 2]$. However, performance starts to
decline for some datasets when it exceeds 2 and 2.5. This indicates
that within an appropriate range, introducing $\gamma$ can effectively
weight the samples, and this effect improves with the increase
of $\gamma$, up to a point where it begins to decrease. This phenomenon
validates the rationality and effectiveness of our designed boundary-aware OE loss.

## 6 CONCLUSION

In this work, we investigate the enhancement of OOD detection
performance on graph-level data through hybrid graph outlier exposure. We demonstrate that not only external outliers are crucial
for graph OOD detection, but internal outliers also play a significant role. Based on this, we propose a carefully designed ID-mixup
based method for synthesizing internal outliers by generating OOD
samples between in-distribution subgroups and aligning these with
external outlier features. Upon obtaining these synthesized internal
outliers, we effectively utilize the characteristics of outlier samples
by adaptively allocating learning weights to OE samples using a
well-designed boundary-aware OE loss. Our HGOE framework provide substantial performance improvements for other graph OOD
detection methods. Extensive experiments on 8 real-world datasets
show its superior performance.

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
