# OpenReview forum: "HGOE: Hybrid External and Internal Graph Outlier Exposure for Graph Out-of-Distribution Detection"
_acmmm.org/ACMMM/2024/Conference — MM2024 Poster_

### Official Review · Reviewer_Bwog · 2024-05-22

**Rating:** 3
**Confidence:** 3

**Summary:**

The paper introduces a novel method named HGOE (Hybrid Graph Outlier Exposure) for improving graph out-of-distribution (OOD) detection. This method integrates external and internal graph data to synthesize outliers, enhancing the detection of OOD instances in graph-structured data. The proposed boundary-aware outlier exposure (OE) loss is designed to optimally use high-quality OOD samples while minimizing the impact of lower-quality ones. The method is demonstrated to be effective across eight real-world datasets, improving the performance of existing OOD detection models.

**Strengths:**

1. Methodological Rigor in Outlier Synthesis: The ID-mixup strategy for generating internal outliers is particularly innovative. This approach not only enhances the diversity of training data but also tailors the outlier generation process to the specific structure of graph data, maintaining the intrinsic properties of the original data distributions.
2. Enhancement of Existing Models: The integration of the HGOE framework with state-of-the-art (SOTA) OOD detection models and its demonstration of improved performance is a significant strength. It shows that HGOE can be adapted to enhance the capabilities of existing models without requiring extensive modifications to their architectures.
3.  Comprehensive Evaluation Across Diverse Domains: The paper conducts extensive experiments across eight real-world datasets from different domains, showcasing the versatility and adaptability of the HGOE framework. This broad evaluation significantly enhances the paper's impact, demonstrating its applicability in diverse real-world scenarios.

**Limitations:**

1. Scalability and Efficiency Concerns: The method involves complex processes like graphon estimation and ID-mixup for outlier synthesis. These processes can be computationally intensive, especially for large-scale graphs, but the paper lacks a detailed discussion on scalability and computational efficiency.
2. Potential Bias in Internal Outlier Generation: The method for generating internal outliers could introduce biases, particularly if the graphon mixup does not accurately reflect the real-world transitions between graph subgroups. This could lead to misleading training signals and affect the model’s performance.
3. Ambiguity in Parameter Selection: The method involves several hyperparameters, such as the balancing hyperparameter in the ID-mixup and the parameters in the boundary-aware OE loss. The sensitivity of the model's performance to these parameters is not extensively discussed, which may pose challenges in tuning the model for optimal performance.

**Suitability:**

2

---

### Official Review · Reviewer_TSWe · 2024-05-25

**Rating:** 4
**Confidence:** 3

**Summary:**

This paper proposes the HGOE method to address the challenge of OOD detection in graph data. The framework integrates external outliers from various domains and synthesizes internal outliers within in-distribution subgroups to improve graph OOD detection performance. The authors introduce a boundary-aware outlier exposure loss that adaptively assigns weights to outliers, maximizing the use of high-quality OOD samples while minimizing the impact of low-quality ones. Experimental results on eight real-world datasets demonstrate the effectiveness of HGOE.

**Strengths:**

1. This paper is well-written, and the technical details are well-elaborated.

2. This paper introduces a hybrid approach that combines external and internal outliers for graph OOD detection. The use of graphon-based ID-mixup to generate internal outliers is novel and addresses the challenge of diversity in graph data.

3. Extensive experimental validation on eight real-world datasets consistently shows the effectiveness of the proposed HGOE framework.

4. The experiment includes detailed visualizations, such as t-SNE and score distributions, to illustrate the effectiveness of the HGOE. These visualizations provide a clear understanding of how the proposed HGOE improves OOD detection.

**Limitations:**

1. The authors need to supplement an algorithmic flow to intuitively show the reader its training process.

2. Lack of theoretical analysis (e.g., time and space complexity) and empirical analysis (e.g., running time) of the computational overhead of the proposed method.

3.  Lack of comparison with other baseline methods in the visualization experiments.

4. How does the hyperparameter $\beta$ affect the HGOE's performance? The reviewer doesn't see the relevant experiments and analyses.

5. Lack of source code of HGOE to guarantee reproducibility.

**Suitability:**

2

---

### Official Review · Reviewer_rs6s · 2024-06-01

**Rating:** 3
**Confidence:** 2

**Summary:**

The paper proposes a novel Hybrid External and Internal Graph Outlier Exposure (HGOE) framework to improve out-of-distribution (OOD) detection performance for graph data using deep graph learning, by incorporating realistic external graph data from various domains and synthesizing internal outliers within in-distribution subgroups, along with a boundary-aware OE loss that intelligently weights outliers to maximize the use of high-quality OOD samples while minimizing the impact of low-quality ones, with the HGOE framework being model-agnostic and designed to enhance existing graph OOD detection models, showing significant performance improvements across 8 real datasets.

**Strengths:**

An ID-mixup method based on graphons to effectively synthesize internal outlier graph samples between in-distribution subgroups.
Introduction of a novel boundary-aware loss function.
Integrating the framework with a state-of-the-art detector, it surpasses competitors across 8 real-world graph datasets, demonstrating superior performance.

**Limitations:**

To facilitate a more comprehensive evaluation, it is suggested that the authors extend the experiments to include the remaining 2 datasets in addition to the current 8, and compare against the GOOD method. Expanding the evaluation scope can provide a more complete picture of the proposed model's capabilities and allow for a fairer comparison.

It would be beneficial to include the publication years of the compared methods in the results tables. This additional information would help readers understand the chronological context and technological advancements of the different approaches.

The authors may consider including the anomaly detection task results from the GOOD method and comparing them with the proposed model. Evaluating on this related task could offer further insights into the strengths and limitations of the proposed approach from an alternative perspective, enriching the analysis.

**Suitability:**

2

---

### Meta-Review · Area_Chair_kwP7 · 2024-06-29

**Recommendation:** Accept (Poster)
**Confidence:** 4

**Metareview:**

This paper received three positive scores (ba,ba,ba) from all of reviewers after rebuttal. The authors have resolved the main concerns raised by reviewers. I am happy to recommend to accept this paper. Please carefully revise the final manuscript according to the comments and discussions.